# Inapparent Strengthening Effect of Twin Interface in Cu/Pd Multilayered Films with a Large Lattice Mismatch

**DOI:** 10.3390/nano9121778

**Published:** 2019-12-13

**Authors:** Shayuan Weng, Xiang Chen, Xing Yue, Tao Fu, Xianghe Peng

**Affiliations:** 1Department of Engineering Mechanics, Chongqing University, Chongqing 400044, China; shayweng@foxmail.com (S.W.); chenxiang@cqupt.edu.cn (X.C.); 20153102014@cqu.edu.cn (X.Y.); 2Advanced Manufacturing Engineering, Chongqing University of Posts and Telecommunications, Chongqing 400065, China; 3State Key Laboratory of Coal Mine Disaster Dynamics and Control, Chongqing University, Chongqing 400044, China

**Keywords:** twin interface, interfacial structure, modulation period, molecular dynamics

## Abstract

It has been found that there are two kinds of interfaces in a Cu/Pd multilayered film, namely, cube-on-cube and twin. However, the effects of the interfacial structure and modulation period on the mechanical properties of a Cu/Pd multilayered film remain unclear. In this work, molecular dynamics simulations of Cu/Pd multilayered film with different interfaces and modulation periods under in-plane tension are performed to investigate the effects of the interfacial structure and modulation period. The interface misfit dislocation net exhibits a periodic triangular distribution, while the residual internal stress can be released through the bending of dislocation lines. With the increase of the modulation period, the maximum stress shows an upward trend, while the flow stress declines. It was found that the maximum stress and flow stress of the sample with a cube-on-cube interface is higher than that of the sample with a twin interface, which is different from the traditional cognition. This unusual phenomenon is mainly attributed to the discontinuity and unevenness of the twin boundaries caused by the extremely severe lattice mismatch.

## 1. Introduction

Nanostructured metallic multilayered films have attracted much attention due to their excellent mechanical properties [1,2,3]. Therefore, the relationship between their mechanical properties and microstructure parameters, and the optimization of these properties, are the hottest topics in this context of recent years [3,4,5,6]. Tremendous efforts have been made to find out the factors influencing the mechanical properties of multilayered films [7,8,9,10,11]. It has been demonstrated that the modulation parameters (such as the modulation period and modulation ratio) and the interfacial structure are the two main factors affecting the mechanical properties of a multilayered film with a particular combination of constituents [12]. Copper-palladium (Cu/Pd) and gold–nickel (Au/Ni) multilayered films are the earliest found multilayered films having excellent mechanical properties [13]. Yang et al. found that the biaxial elastic moduli, Y [111], of Cu/Pd and Au/Ni multilayered films, increases drastically from 0.27 to 1.31 TPa and from 0.21 to 0.46 TPa, respectively [13]. Subsequently, the shear and Young’s modulus of the Cu/Pd multilayered film does not show significant improvements [14], which raises the question of whether the supermodulus effect exists in the Cu/Pd multilayers. This dispute is hard to resolve by experiments alone, because of the variations in the preparation and test conditions [15]. The enhancement of elastic properties may be controversial, but the increase of strength and hardness in Cu/Pd multilayered film is unquestionable. Since Cu/Pd multilayered films were first found to have excellent mechanical performance, such as an increase of strength and hardness, in this work, it will be selected as a representative to study the mechanical properties of multilayered films [16,17,18,19,20].

The modulation period (*λ*) and modulation ratio (*η*) are the thickness of a representative unit in a multilayered film and the ratio of the layer thickness of each constituent, respectively [21,22]. The change of *λ* and *η* may induce a change of interface proportion and the space for dislocation movement. Moreover, the grain boundary-dominated deformation mechanism may also vary with the variation of *λ* and *η*. For example, as *λ* decreases, the dominant deformation mechanisms in metallic multilayered films may vary from “dislocation pile-up” to “confined layer slip”, or to “interface crossing” [21]. With the decrease of *λ*, the strengthening caused by the dislocation pile-up is weakened due to fact that a large number of dislocations between the interfaces can be stored in layers. However, the strengthening induced by the glide of single dislocations confined by interfaces (“confined layer slip”) or the weakening resulting from dislocation crossing the interface (“interface crossing”) dominate would play a leading role. Therefore, *λ* and *η* have significant effects on the mechanical properties of a multilayered film. This has an essential guiding significance to establish the relationship between the mechanical performance and modulation parameters for the design and development of high-performance multilayered films with excellent mechanical properties. In this work, we will mainly focus on the effects of *λ* on mechanical properties.

Last but not least, interfaces, the transition zone between two components in nanostructured metallic multilayered films, can act as sources of defects, sinks of defects via absorption and annihilation, barriers to the motion of defects, and storage sites of defects [23], which usually significantly affects the mechanical properties of multilayered films. According to lattice mismatch parameter on an interface, δ = (*a*_A_ − *a*_B_)/*a*_A_, the multilayered films consisting of the constituents (A and B) with the same lattice structure can be divided into three groups, namely, coherent (δ ≤ 5%), semi-coherent (5% ≤ δ ≤ 25%), and non-coherent multilayer (δ ≥ 25%) groups. It is known that molecular dynamics (MD) simulation is an effective method to study the deformation and mechanical behavior of materials [24,25,26,27,28]. For example, with MD simulation, Shao et al. studied the relaxation mechanisms and misfit dislocation patterns of semi-coherent interfaces in a FCC/FCC multilayered film [29,30,31,32], where FCC is the abbreviation for Face-Centered Cubic. Using MD simulation, Weng et al. investigated the deformation behavior of Cu/Ni multilayers with coherent, semi-coherent, and coherent twin interfaces, and their effects on the mechanical properties [33]. These works revealed that the effect of twin boundaries could be stronger than that of other kinds of boundaries. The structure of Pd films on Cu(111) was investigated using medium energy ion scattering, and two kinds of common interfaces, namely, the twin interface and cube-on-cube interface, were observed in the FCC/FCC multilayer [3]. The effects of these two interfacial structures on the mechanical properties of Cu/Pd multilayered films are still unclear. Moreover, the Cu/Pd multilayered films have a much larger lattice mismatch (~7.07%) than the Cu/Ni multilayered film (~2.62%), which may result in less significant strengthening. Therefore, it is necessary to study whether the conclusions obtained from the low mismatched multilayered films are applicable to the Cu/Pd multilayered films.

In this work, to investigate the effects of the interfacial structure and modulation period on the mechanical properties of the multilayered films, we conducted a series of MD simulations for the tensile deformation of Cu/Pd multilayered films with cube-on-cube and with twin interfaces, respectively.

## 2. Simulation Details

The second nearest-neighbor modified embedded atom method (2NN MEAM) potential [34,35] was selected to calculate the interatomic forces. The parameters of the 2NN MEAM potentials for the two single elements (Cu-Cu and Pd-Pd) are given in Table 1 [36]. To uncover the formation mechanism of growth twins, we developed a set of 2NN MEAM potential parameters for Cu/Pd system in our previous work, which can not only describe the basic mechanical properties of both pure Pd and Cu, as well as their alloys, but also reproduce the evolution of growth twins [15]. The atomic interaction between Cu and Pd is described with a binary 2NN MEAM potential, with the parameters listed in Table 2 [15].

The multilayered films are prone to grow along the <111> direction [37,38], and twin interfaces are often Cu{111}/Pd{111}. Hence, we only consider the interfaces between the Cu{111} and Pd{111} in this work. Figure 1a,b show the atomic model of the sample with a cube-on-cube and twin interface, abbreviated as COC and Twin herein for brevity, respectively. Since the periodic boundary conditions were used to improve the computational efficiency, the performance of a multilayered film can be represented with a model containing one representative cell, as shown in Figure 1. For Cu, in the COC and Twin samples, the *x-*, *y-* and *z*-axes correspond to crystallographic orientations of $\left[01\bar{1}\right]$, $\left[\bar{2}11\right]$ and [111], respectively. For Pd, in the COC sample, the *x-*, *y-*, and *z*-axes correspond to the identical crystallographic orientations as that in Cu, however, those in the Twin sample correspond to crystallographic orientations of $\left[01\bar{1}\right]$, $\left[\bar{2}11\right]$ and $\left[\bar{1}\bar{1}\bar{1}\right]$. The crystal orientation of the two types of interfaces can also be seen visually in Figure 1a,b. To investigate the effects of *λ*, for each interface, five samples, with *λ* = 39.00 Å, 78.00 Å, 117.00 Å, 156.00 Å, and 234.00 Å, were built, respectively. The modulation ratio, *η*, was set equal to the ratio of the lattice constants of Cu and Pd, 3.615/3.890. The lengths in *x-* and *y*-axes were 219.83 Å and 190.38 Å, respectively, which ensures that the Cu layer and Pd layer have a perfect crystal lattice and are without any in-plane artificial grain boundary. Figure 1c,d show the atomic configurations on the *y*-*z* plane, colored by atomic types.

The dislocation extraction algorithm (DXA) was used to analyze local structures, which can divide the atoms into different types of local structures (Face-Centered Cubic FCC, body-centered cubic BCC, hexagonal close packed HCP, etc.) based on their local environment, and identify the common dislocations in FCC crystal, as well as determine their Burgers vectors and output dislocation lines [39]. The atoms are colored according to their local structures using the following rule: Green for FCC, red for HCP, blue for BCC, and white for “other” local crystal structures, as shown in the inset of Figure 1. The red atoms can be further subdivided into a stacking fault (SF) with two adjacent red layers and a twin boundary (TB) with one single red layer of atoms, respectively. The open-source program, Open Visualization Tool (OVITO), was used to color the atoms with various local lattice structures for visualization [40]. OVITO is a post-processing and scientific data visualization and analysis software, working with molecular and other particle-based data, typically generated in numeric simulation models from materials science, physics, and chemistry disciplines [40]. Figure 1e,f show the atomic configurations on the *y-z* plane, colored by the local structure, where one can see that the atoms in the Cu and Pd layers are distributed in the same way in the COC sample, but are symmetrically distributed about the interface in the Twin sample.

Before loading, energy minimization and relaxation were performed successively to optimize the interface structure and achieve an equilibrium system, respectively. The conjugate gradient (CG) algorithm was used to optimize the interfaces at the temperature of 0 K. Then, the sample was relaxed at 300 K for 20 ps under the isothermal-isobaric (NPT) ensemble to obtain an equilibrium system with zero pressure in *x*-, *y*- and *z*-directions through a Nose–Hoover thermostat [41] and a Nose–Hoover pressure barostat [42]. Nose–Hoover thermostat and Nose–Hoover pressure barostat have been included in the NPT ensemble with *Tdamp* and *Pdamp*, respectively, which determine how rapidly the temperature and pressure is relaxed, respectively [43]. Uniaxial tension simulations were performed by stretching the sample in the *x*-direction at the strain rate of 1 × 10^9^ s^−1^ using the Large-scale Atomic/Molecular Massively Parallel Simulator (LAMMPS) [43]. LAMMPS is an open-source molecular dynamics program, with a focus on materials modeling, and has the potential for solid-state materials (metals, semiconductors), soft matter (biomolecules, polymers), and coarse-grained or mesoscopic systems [43]. During loading, the pressures in *y*- and *z*-directions were kept at zero to satisfy the requirement of uniaxial tensile deformation. In all the simulations, periodic boundary conditions were used in *x-*, *y-*, and *z*-directions, respectively.

## 3. Results and Discussions

### 3.1. Characterization of Interfaces

Figure 2 shows the evolution of the interfacial microstructure of the samples with *λ* = 78 Å, where the atoms are colored with local structures, and the atoms identified as an FCC structure have been removed for clarity. Figure 2a,b show the interfacial structure after energy minimization, where one can see that lattice mismatches in the COC and Twin samples are accommodated by triangular 1/6<112> partial dislocation networks. These networks are composed of three kinds of misfit dislocations, with Burgers vectors of $\frac{1}{6}[\bar{2}11]$, $\frac{1}{6}[\bar{1}\bar{1}2]$, and $\frac{1}{6}[\bar{1}2\bar{1}]$, respectively. The intersection of these dislocation lines has an angle of 60° and forms dislocation nodes, as shown in Figure 2a,b. The dislocation lines with the identical Burgers vector are parallel with each other, and the distance between two adjacent dislocation lines is about 30.87 Å.

On the other hand, the difference we have found lie in the form of atomic stacking within the triangle. The red atoms in the COC sample make up the SF, but those in the Twin sample form a twin interface, which can be confirmed by the atomic configurations on the *y*-*z* plane, as shown in Figure 1e,f. In Figure 2, the atoms identified as having a FCC structure have been removed for clarity, and the red atoms constitute SF planes or TBs. In the COC sample, one can see that there are no non-FCC atoms in the white triangular regions that can be regarded as coherent regions (CRs). The interface in the COC sample is composed of triangular SF regions, CRs, dislocation lines, and dislocation nodes. In the Twin sample, we can see that there are one-layer red atoms in all triangular regions, and the atoms in the adjacent triangle are not in the same atomic layer. These one-layer red atoms form TB. From Figure 1f, we can find that the TBs of adjacent triangles consist of Cu and Pd atoms, respectively. Therefore, the interface of the Twin sample is composed of triangular TBs (consisting of Cu or Pd atoms), dislocation lines, and dislocation nodes.

The dislocation lines remain straight after energy minimization, as shown in Figure 2a,b, but there is residual stress in the system, especially in the in-plane directions. During the energy minimization, the energy of the system is minimized by the slight movement of atoms. The size of samples in each direction cannot change freely. This process is mainly to optimize the local structure, specifically the interfacial structure here. Hence, due to the limitations of the sample size, there are still some residual stresses in all directions after energy minimization. To relax the internal residual stress and obtain an equilibrium system, the samples were relaxed at 300 K for 20 ps. In the relaxation, the sample size allows changing to relax the stress in all directions to zero pressure. Figure 2b,c show the interface microstructures after relaxation for 10 ps and 20 ps, respectively, where the dislocation lines are bent gradually, and the shapes of the SF and TB change synchronously. Figure 3a shows the variations of the atomic fractions of different local structures against relaxation time (*T*), where the proportion of the atoms with different local structures hardly change with the increase of *T*, indicating that the areas of the SFs and of TBs do not change significantly. However, the total lengths of the dislocation lines change obviously, as shown in Figure 3b. With the increase of *T*, the lengths of dislocation lines in the COC and Twin samples increase and gradually tend to asymptotic values, indicating that the relaxation of the internal residual stress is achieved with the increase of the length of the dislocation lines. The stability of the dislocation line length indicates that the system has reached an equilibrium state. The total lengths of the dislocation lines (*L*) of the two initial samples are almost identical, but as the relaxation continues, *L* of the Twin sample develops faster than that of the COC sample.

### 3.2. Deformation Behaviors

Figure 4a shows the stress-strain (*σ-ε*) curves of the samples, where *λ* = 78 Å. These curves can be divided into three stages, marked with (I), (II), and (III), respectively. In stage (I), the curves develop almost linearly to their peaks. Figure 5a,b show the microstructures of the samples COC and Twin at *ε* = 0.05, where the atoms identified as “FCC” and “other” structures have been removed for a clearer observation. It can be seen that, besides the atoms in a FCC structure, there are no other atoms or dislocations in the space between the interfaces, which indicates that elastic deformation mainly occurs in this stage. Using the DXA algorithm, we can classify atoms in different patterns of lattices based on local microstructures, and then output the number of atoms in different local structures, with which the fractions of the atoms in different patterns of lattices can be calculated. Figure 4b shows the atomic fractions of both the FCC and HCP local structures as a function of strain, where they do not change significantly in stage (I). The lengths of the 1/6<112> PD lines and the total length of the dislocation lines hardly varies with the increase of strain in stage I, as shown in Figure 4c. In this stage, the length of 1/6<110> stair rod dislocations, which usually form with the reaction of two PDs, is zero, indicating that no dislocation reaction has occured. In short, there is no phase transition, dislocation nucleation, and dislocation reactions in this stage, implying elastic deformation in this stage. The slopes of the *σ-ε* curves and Young’s modulus (*E*) in this region are nearly identical, which indicates that the interface structures have insignificant effects on the elastic properties. However, the maximum stress and the corresponding strain of the COC sample (8.39 GPa and 0.067, respectively) are both larger than that of the Twin sample (7.98 GPa and 0.064, respectively). The corresponding microstructures are presented in Figure 5c,d, respectively, where one can see that the several SFs bounded by partial dislocations (PDs) form in the Pd layer. The top views of the interface structures, corresponding to Figure 5c,d, are shown in Figure 6a,b, respectively, where PDs nucleate from the misfit dislocation network due to the higher local mismatch stress and slip on the {111} plane, leaving SFs behind the Pd layer. The SFs form firstly in the Pd layer, which could be ascribed to the lower stacking fault energy of Pd than that of Cu (8~18 mJ/m^2^ for Pd, 20~30 mJ/m^2^ for Cu) [44,45].

With increasing strain in stage (II), some dislocations glide into the Cu layer, which can be verified by the microstructures in the COC and Twin samples at *ε* = 0.1, as shown in Figure 5e,f, respectively. In this stage, many atomic local structures change from FCC to HCP (Figure 4b), and the lengths of the 1/6<112> PD lines and the total length of the dislocation lines increase significantly (Figure 4c). Some 1/6<110> stair rod dislocations form, and their lengths increase with the increase of strain. The sharp drops of stress (Figure 4a) can be ascribed to the synergistic effect between the nucleation and glide of dislocations, as well as the dislocation interaction. However, in the latter part of stage (II), the curves tend to be parallel, but still fluctuate, indicating that a dislocation reaction has occurred, though it insignificantly affects the number of dislocations. The difference between the two kinds of interfaces is that in the COC sample, the SFs and dislocation lines are mainly parallel (Figure 5e,g), while in the Twin sample they are mainly symmetric (Figure 5f,h).

In stage (III), the stress tends to fluctuate around a constant with the increase of strain, which can be regarded as the flow stress stage. In this stage, the atomic fractions of the FCC and HCP structures change insignificantly with the increase of strain, but the lengths of dislocation lines change obviously, which shows that the interaction between dislocations does not significantly change the area of the SFs and TBs. It can be found from microstructure analyses that, besides SFs (two adjacent red layers atoms), some TBs (a single layer of red atoms) are generated through the layer-by-layer movement of 1/6<112> partial dislocation on the adjacent {111} planes [46,47]. Three sets of dislocation lines nucleated from the mismatched dislocation network (Figure 2) glide on the {111} planes, which provides favorable conditions for twinning. Therefore, slight fluctuations in the *σ-ε* curve can be attributed to the effect of the competition between the nucleation of new dislocations, the reactions between dislocations, and twinning. The reaction of dislocations dominates the deformation in stage (III), where the reactions are mainly the decomposition of <110> perfect dislocations into <112> PDs, or the synthesis reaction of <112> PDs into <110> perfect dislocations. Therefore, the numbers of the two kinds of dislocations change reversibly. From Figure 4b, one can see that the atomic fraction of HCP is lower than that of FCC, which implies that dislocation nucleation and slip-induced microstructural changes are not sufficient to convert all FCC structures to HCP structures. It is interesting to find that the flow stress of the COC sample is higher than that of the Twin sample, which is different from the traditional cognition that twin boundary has a strengthening effect.

### 3.3. Effects of the Modulation Period

Figure 7 shows the *σ-ε* curves of the COC and Twin samples with different modulation periods (*λ*s), where the curves have similar shapes to those shown in Figure 4a. To explore the effects of *λ*, the variations of the Young’s modulus (*E*), maximum stress (*σ*_m_), and flow stress (*σ*_f_) against *λ* obtained from the *σ-ε* curves in Figure 7 are shown in Figure 8a–c, respectively. *E* is obtained by fitting the slope of these curves in the strain range of 0.0–0.05. For the samples with different interfacial structures and different *λ*s, *E* lies between 128 and 131 GPa (Figure 8a), indicating that the modulation period and interfacial structure have an insignificant effect on the elastic property. This can further confirm that the so-called supermodulus in multilayers may not be universal.

With the increase of *λ*, the peak stress, *σ*_m_, shows an upward trend, which is similar to that of Cu/Ta [12]. *σ*_m_ corresponds to the nucleation of dislocation from the interface (Figure 6). Interface density increases with the decrease of *λ*, which, in turn, would increase the probability of dislocation nucleation. As the interface density increases, the required nucleation stress may decrease [12]. If *λ* is small, it may only need small stress to activate dislocation nucleation from the interface. We can also see that the *σ*_m_ of the COC sample is generally higher than that of the Twin sample, which can be attributed to the following two reasons: (1) The interface in the COC sample is more stable than that that in the Twin sample, and (2) the twin interface is discontinuous and uneven (step). Figure 8d shows the difference between the interfacial energy density of the Twin sample and that of the COC sample, $\Delta E(\lambda )=\left(E_{\mathrm{Twin}}-E_{\mathrm{COC}}\right)/2A$, where *E*_COC_ and *E*_Twin_ are the potential energies of the COC and Twin samples, with different *λ* values, respectively, and *A* is the area of the interfaces. It can be seen that, for different *λ* values, Δ*E* is a positive value, which indicates that the interface in the COC sample is more stable than that in the Twin sample. However, the difference in *σ*_m_ is still significant, even when Δ*E* decreases to almost zero with the increase of *λ*. Therefore, the discontinuity and steps of the interface in the Twin sample (Figure 6b) may account for its lower *σ*_m_. *σ*_f_ is the average stress of the curves in 0.1 < *ε* < 0.175, as can be seen in Figure 8c. In this stage, the dislocation interaction and dislocation slip impeded by the interface are the main deformation behaviors, and the confined layer slip (CLS) [48,49,50,51], caused by the dislocation slip being impeded by the interface, is the primary strengthening mechanism. With increasing *λ*, this strengthening effect is weakened, resulting in a decrease of *σ*_f_. Besides, the *σ*_f_ of the COC sample is higher than that of the Twin sample, which may overturn the existing cognition that the strengthening effect of TB is noticeable. Similar to *σ*_m_, this phenomenon can also be attributed to the discontinuity and unevenness (step) of the twin interface, as shown in Figure 6b.

## 4. Conclusions

In this work, we performed the MD simulation of the in-plane tension of Cu/Pd multilayers with different kinds of interfaces and modulation periods to investigate the effects of the interfacial structure and modulation period. We found that the interfacial misfit dislocation net exhibits a periodic distribution of triangles. The interface of the COC sample (the sample with a cube-on-cube interface) consists of a triangular stacking fault region, coherent region, and dislocation lines and their intersections (nodes), while that of the Twin sample (the sample with a TB interface) consists of TBs consisting of Cu or Pd atoms, partial dislocation lines, and dislocation nodes. The residual internal stress is released through the bending of dislocation lines. With the increase of the modulation period, the maximum stress exhibits an upward trend, while the flow stress exhibits a downward trend. It is interesting to find that the flow stress of the COC sample is larger than that of the Twin sample, which is different from the conventional cognition that the twin boundary has a strengthening effect. This unusual phenomenon is mainly attributed to the discontinuity and unevenness of the twin boundaries caused by the severe mismatch between the lattice parameters.

## Figures and Tables

**Figure 1 nanomaterials-09-01778-f001:**
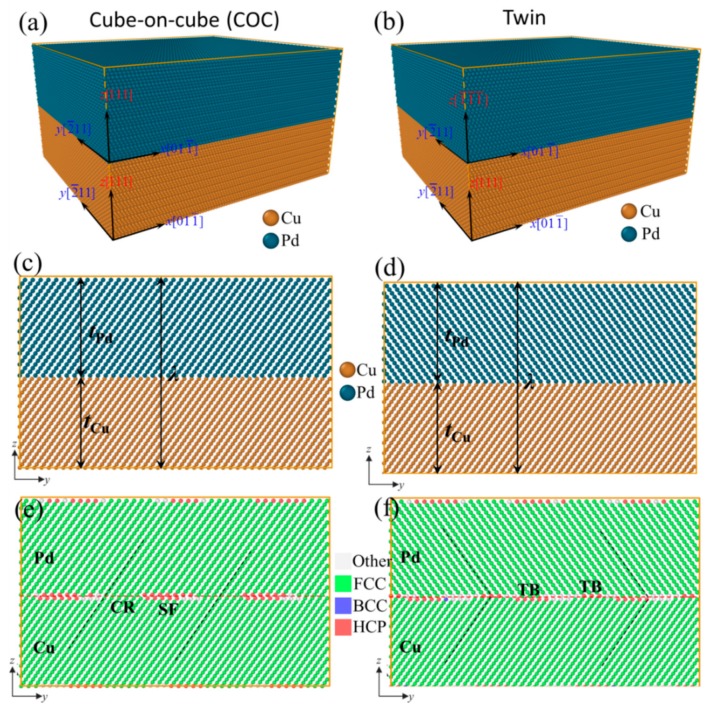
(**a**,**b**) Atomic model of samples on the cube-on-cube interface (COC) and twin interface (Twin). (**c**,**d**) and (**e**,**f**) denote the atomic configurations in the *y*-*z* plane, colored by atomic types and local structure, respectively.

**Figure 2 nanomaterials-09-01778-f002:**
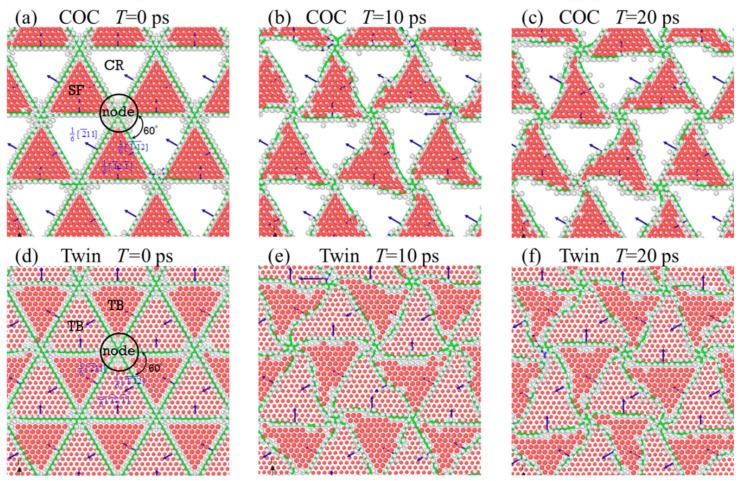
Top views of the interfacial microstructure in the (**a**–**c**) COC and (**d**–**f**) Twin samples, with λ = 78 Å at different relaxation times (*T*), with green lines denoting dislocation lines.

**Figure 3 nanomaterials-09-01778-f003:**
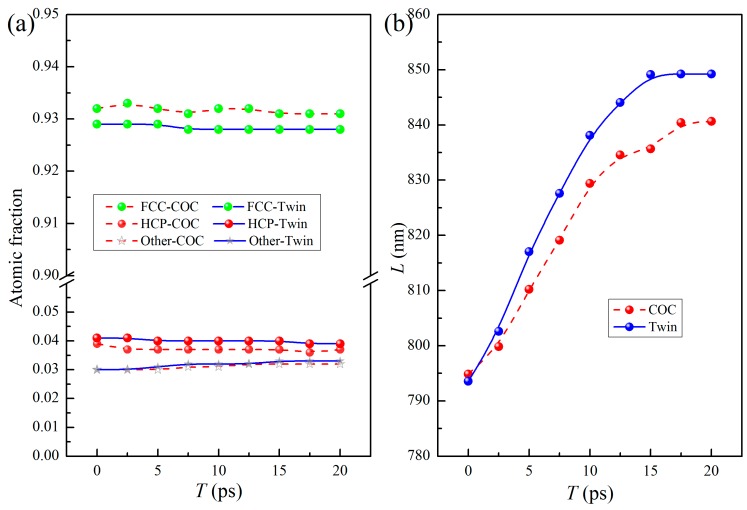
(**a**) Variations of atomic fractions of different local structures against time *T*. (**b**) Variations of dislocation line lengths (*L*) against time *T*.

**Figure 4 nanomaterials-09-01778-f004:**
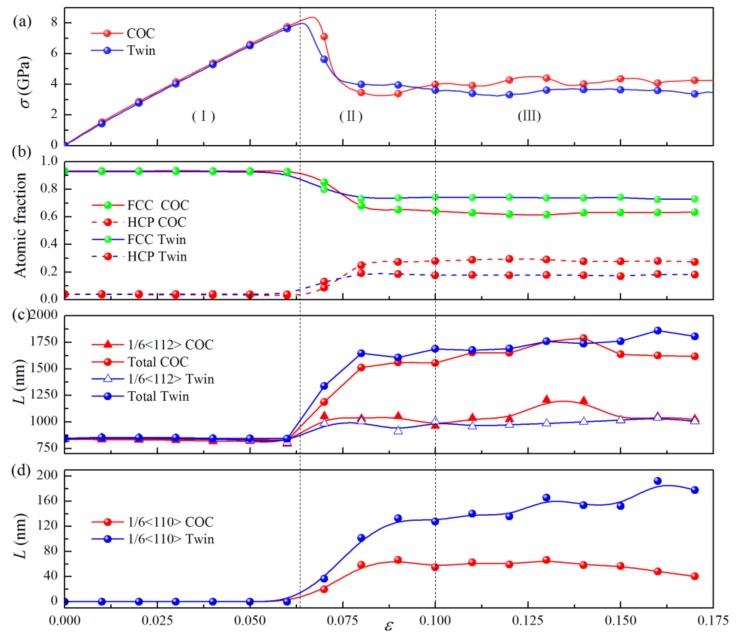
(**a**) *σ-ε* curves for the COC and Twin samples, with λ = 78 Å. (**b**) Atomic fractions of the FCC and HCP local structures as a function of strain. (**c**,**d**) Lengths of dislocation lines as a function of strain.

**Figure 5 nanomaterials-09-01778-f005:**
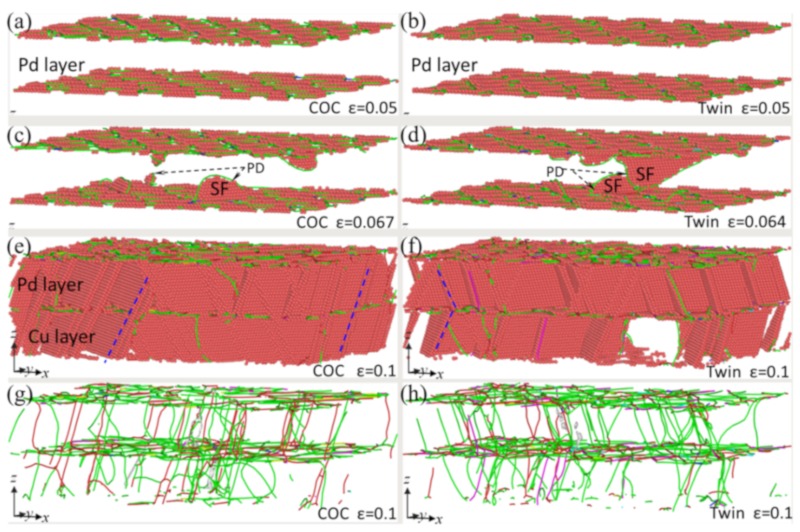
Evolutions of microstructures in COC (left column) and Twin (right column) samples under different tensile strains. (**a**) COC at *ε* = 0.05; (**b**) Twin at *ε* = 0.05; (**c**) COC at *ε* = 0.067; (**d**) Twin at *ε* = 0.064; (**e**) COC at *ε* = 0.1; and (**f**) Twin at *ε* = 0.1. (**g**) and (**h**) are the distributions of dislocation line of COC and Twin at *ε* = 0.1, respectively.

**Figure 6 nanomaterials-09-01778-f006:**
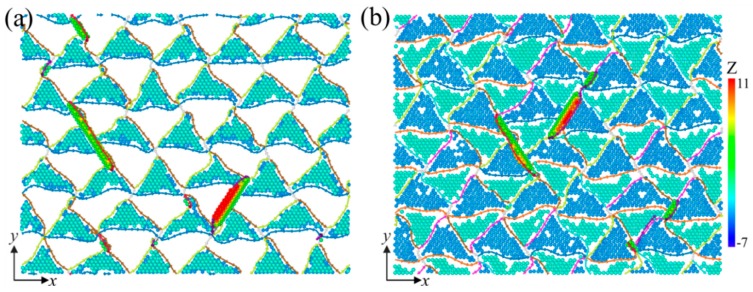
Top views of interfacial microstructures. (**a**) COC sample at *ε* = 0.067. (**b**) Twin sample at *ε* = 0.064, with atoms colored by their Z values.

**Figure 7 nanomaterials-09-01778-f007:**
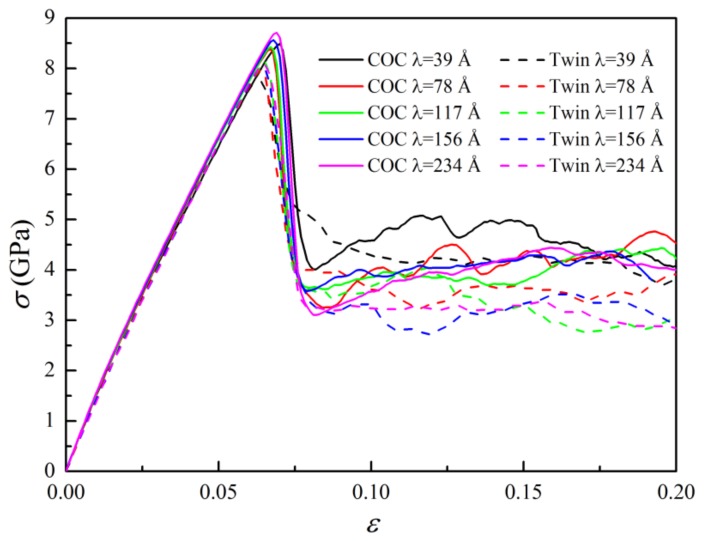
*σ-ε* curves of the COC and Twin samples with different *λ*s values.

**Figure 8 nanomaterials-09-01778-f008:**
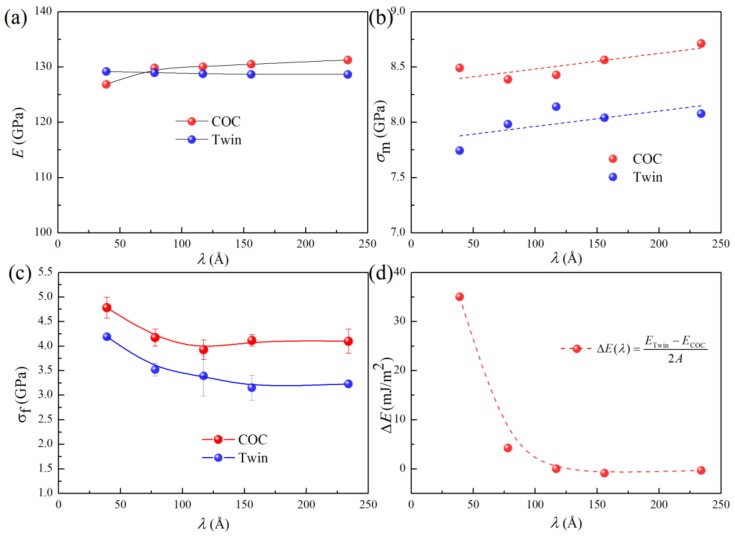
(**a**–**c**) *E*, *σ*_m_, and *σ*_f_ as a function of *λ,* respectively. (**d**) Interfacial energy difference as a function of *λ.*

**Table 1 nanomaterials-09-01778-t001:** Second nearest-neighbor modified embedded atom method (2NN MEAM) potential parameters for pure Cu and Pd [30], with *E_c_* (eV), *r_e_* (Å), *α*, *A*, *β*^(0~3)^, *t*^(0~3)^, *C*, and *d* denoting cohesive energy, equilibrium nearest-neighbor distance, exponential decay factor, scaling factor for the embedding energy, exponential decay factor, weight factor for the atomic densities, screening parameter, and the adjustable parameter, respectively.

Element	*E_c_* (eV)	*r_e_* (Å)	*α*	*A*	*β* ^(0)^	*β* ^(1)^	*β* ^(2)^	*β* ^(3)^	*t* ^(0)^	*t* ^(1)^	*t* ^(2)^	*t* ^(3)^	*C* _min_	*C* _max_	*d*
Cu	3.54	2.555	5.15	0.94	3.83	1.00	6.0	2.2	1.0	2.72	3.04	1.95	1.21	2.8	0.05
Pd	3.91	2.750	6.42	0.94	2.75	5.15	6.0	2.2	1.0	4.50	1.47	4.85	1.69	2.8	0.05

**Table 2 nanomaterials-09-01778-t002:** 2NN MEAM potential parameters for a Cu-Pd system [15]. *E*_c_, *r*_e,_ and *B* are cohesive energy, equilibrium nearest-neighbor distance, and bulk modulus, respectively.

Parameter	*E_c_* (eV)	*r_e_* (Å)	*B* (GPa)	*d*	Cu–Pd–Cu		Pd–Cu–Pd		Cu–Cu–Pd		Pd–Pd–Cu	
					*C* _min_	*C* _max_	*C* _min_	*C* _max_	*C* _min_	*C* _max_	*C* _min_	*C* _max_
Value	3.725	2.593	106.2	0.05	0.65	1.44	0.78	1.44	1.44	2.8	1.44	2.8

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
