# Peer review of "Inapparent Strengthening Effect of Twin Interface in Cu/Pd Multilayered Films with a Large Lattice Mismatch"

_nanomaterials, 2019, doi:10.3390/nano9121778_

Round 1

Reviewer 1 Report

The manuscript titled “Inapparent strengthening effect of twin interface in 2 Cu/Pd multilayered films with a large lattice mismatch” has a message for the readers. Even though, there are many points need to be improved to publish in this journal. There are many points don’t exhibit clear message for a reader. Hence, after the effective modification the manuscript is recommended for the publication. There are typos and mistakes which need to be corrected with proper way.

Line 45-46 “have excellent performance” explain what type of performance. Line 54. include more explanation about “dislocation pile-up” to “confined layer slip”, Line 63-64, “coherent multilayer, 63 semi-coherent, and non-coherent according to the lattice mismatch” explain how to find these parameters. Line 82. What type of interatomic forces were used and how did you used it for the calculations? In table 1 and 2, there is no clear explanation for 0-3 both βand t. in addition, the t made confusion in fig. 3(a). add clear explanation and differentiate properly. Line 121 include more details of FCC, BCC, HCP and explain denote it in figures 1 and 2 where relevant Line 122 the Burgers vector include more explanation and calculations, Line 125-126, explain more details-how to find in a crystal structure Stacking fault (SF) with two adjacent red layers, and twin boundary (TB) Line 127 and 138, what is OVITO and LAMMPS code and explain more details. Line 128, In figure 1(e-f), why Cu and Pd exhibited in same color Line 136, explain more details about Nose-Hoover thermostat. Line 158-159, “ dislocation lines has an angle of 60° and forms a dislocation node” indicate this in proper location Line 161-162, “sample Twin there are red atoms in all triangles, but in the sample COC there are no atoms in the 161 triangle regions, indicating that in these regions the interfaces are coherent” the sentence is not clear. Rewrite in details. Line 163, “triangular dislocation region” indicate it in fig. Line161-167, sample Twin there are red atoms in all triangles, but in the sample COC there are no atoms in the 161 triangle regions, indicating that in these regions the interfaces are coherent. Therefore, the interface in the sample COC can be regarded to consist of a triangular dislocation region, coherent region, and dislocation lines as well as their intersections (nodes). In the sample Twin, the remaining red atoms in the triangle form TBs, and the red atoms in the adjacent triangle are not in the same atomic layer, as shown in Fig. 1(f). Therefore, the interface of the sample Twin can be regarded as TBs consisting of Cu and Pd atoms, partial dislocation lines, and intersection node” the paragraph is not clear. Please re-write it. Table and figure should be in bold. Line 173-175, “The dislocation lines are straight after energy minimization, as shown in Figs. 2(a) and (d), but there is residual stress in the system, especially in the in-plane directions. To relax the internal residual stress and obtain an equilibrium system, samples are relaxed at 300 K for 20 ps.” Explain why. Line 178 “atomic fractions of different local” how to find it. Explain in detail. Figure 4a, why бdropped in stage 2 correlate this with stage I and stage III. Figure 4b, why atomic fraction of HCP is lower than that of FCC and what is the reason for the change in stage 2 and parallel behavior Figure 4c. why there is no change in stage 3 and 1/6<112> exhibited lower value COC and twin Figure 4d. the 1/6<110> twin displays igher value while 1/6<112> shows lower value. What is the reason for and why <110> shows lower value than <112> What is the different between fig 4a and 7. Author is advised to include reflective planes of <110 or112> in figure 8. Make conclusion in to a paragraph.

Author Response

Response to Reviewer 1 Comments

Point 1: The manuscript titled “Inapparent strengthening effect of twin interface in Cu/Pd multilayered films with a large lattice mismatch” has a message for the readers. Even though, there are many points need to be improved to publish in this journal. There are many points don’t exhibit clear message for a reader. Hence, after the effective modification the manuscript is recommended for the publication. There are typos and mistakes which need to be corrected with proper way.

Response 1: We thank the reviewer very much for valuing our manuscript and for his/her recommendation of our manuscript in the possible publication in Nanomaterials. We have carefully checked the presentation and tried our best to eliminate the grammatical and spell mistakes in the revised manuscript.

Point 2: Line 45-46 “have excellent performance” explain what type of performance.

Response 2: Thank the reviewer for this comment. The “excellent performance” has been described in Line 37-40 of the initial manuscript; however, to highlight, more details have been added in the revised manuscript.

Point 3: Line 54. include more explanation about “dislocation pile-up” to “confined layer slip”.

Response 3: Thank the reviewer for this kind suggestion. More descriptions about “dislocation pile-up” and “confined layer slip” have been added in the revised manuscript.

Point 4: Line 63-64, “coherent multilayer, semi-coherent, and non-coherent according to the lattice mismatch” explain how to find these parameters.

Response 4: Thank the reviewer for this kind comment. The definition of the lattice mismatch and the method to classify the multilayer films based on the lattice mismatch have been added in the revised manuscript.

Point 5:  Line 82. What type of interatomic forces were used and how did you use it for the calculations?

Response 5: Thank the reviewer for this comment. It is known that the interatomic potential gives the relationship between the potential energy and the distance between atoms. According to this relationship, the interatomic interaction force can be obtained by considering the potential energy between the atom and surrounding atoms and then deriving the distance. This is the basis of molecular dynamics; hence the relevant details are not presented in this article.

Point 6:  In table 1 and 2, there is no clear explanation for 0-3 both β and t. in addition, the t made confusion in fig. 3(a). add clear explanation and differentiate properly.

Response 6: Thank the reviewer for this comment. As described in the title of Table 1, β(0-3) and t(0-3) are the exponential decay factor and weight factor for the atomic densities. They are usually adjustable parameters for different systems, which have been described in the references 28-30. The determination process of these parameters is very complicated, and it is not the work of this work. We use the available potential parameters directly; therefore the relevant details are not presented in the manuscript. To facilitate the reader to check the determination process of parameters, we have cited relevant references, Ref. [28-30].

We have also modified the “t” to “T” to avoid confusion.

Point 7:  Line 121 include more details of FCC, BCC, HCP and explain denote it in figures 1 and 2 where relevant. Line 122 the Burgers vector include more explanation and calculations. Line 125-126, explain more details-how to find in a crystal structure Stacking fault (SF) with two adjacent red layers, and twin boundary (TB).

Response 7: Thank the reviewer for this comment. The local structures (FCC, BCC, HCP) and dislocation lines are identified by the dislocation extraction algorithm (DXA), which is proposed by Stukowski and his colleagues. This algorithm has been compiled into OVITO software. The algorithm contains a lot of content; meanwhile, we have not modified this algorithm. Considering the length of the manuscript, some details are not presented in this manuscript. To make it easier for the reader to consult the algorithm, we cite the original references [Refer to Ref. 39 in the revised manuscript].

Point 8:  Line 127 and 138, what is OVITO and LAMMPS code and explain more details.

Response 8: Thank the reviewer for this comment. LAMMPS is a classical molecular dynamics code with a focus on materials modeling. It has the potentials for solid-state materials (metals, semiconductors) and soft matter (biomolecules, polymers) and coarse-grained or mesoscopic systems. And OVITO is a scientific data visualization and analysis software for working with molecular and other particle-based data, typically generated in numeric simulation models from materials science, physics and chemistry discipline. In this work, we use LAMMPS to perform MD simulations and use OVITO to analyze the formation and evolution of microstructures visually.

The introductions of these two codes have been added in the revised manuscript.

Point 9:  Line 128, In figure 1(e-f), why Cu and Pd exhibited in same color

Response 9: Thank the reviewer for this comment. As described in the caption of Fig. 1, the atoms in Figs. 1 (e) and (f) are colored by local structure (lattice structure). Both Cu and Pd are FCC structures; hence they are in the same color.

Point 10:  Line 136, explain more details about Nose-Hoover thermostat.

Response 10: Thank the reviewer for this comment. More details about Nose-Hoover thermostat have been added in the revised manuscript.

Point 11: Line 158-159, “dislocation lines has an angle of 60° and forms a dislocation node” indicate this in the proper location.

Response 11: Thank the reviewer for this kind suggestion. The related marks have been made in Figs. 2 (a) and (d) in the revised manuscript.

Point 12: Line 161-162, “sample Twin there are red atoms in all triangles, but in the sample COC there are no atoms in the triangle regions, indicating that in these regions the interfaces are coherent” the sentence is not clear. Rewrite in detail. Line 161-167, sample Twin there are red atoms in all triangles, but in the sample COC there are no atoms in the triangle regions, indicating that in these regions the interfaces are coherent. Therefore, the interface in the sample COC can be regarded to consist of a triangular dislocation region, coherent region, and dislocation lines as well as their intersections (nodes). In the sample Twin, the remaining red atoms in the triangle form TBs, and the red atoms in the adjacent triangle are not in the same atomic layer, as shown in Fig. 1(f). Therefore, the interface of the sample Twin can be regarded as TBs consisting of Cu and Pd atoms, partial dislocation lines, and intersection node” the paragraph is not clear. Please re-write it.

Response 12: Thank the reviewer for this kind suggestion. We have rewritten this paragraph, and tried our best to make it as clear as possible.

Point 13:  Line 163, “triangular dislocation region” indicate it in fig. 2.

Response 13: Thank the reviewer for this kind suggestion. We are sorry for this mistake. The triangular dislocation region should be the “triangular stacking fault region”, we have modified this mistake in the revised manuscript.

Point 14:  Table and figure should be in bold.

Response 14: Thank the reviewer for this kind suggestion. We have modified the fonts of the captions of tables and figures to meet the requirements.

Point 15:  Line 173-175, “The dislocation lines are straight after energy minimization, as shown in Figs. 2(a) and (d), but there is residual stress in the system, especially in the in-plane directions. To relax the internal residual stress and obtain an equilibrium system, samples are relaxed at 300 K for 20 ps.” Explain why.

Response 15: Thank the reviewer for this kind comment. During the energy minimization, the energy of the system is minimized by the slight movement of atoms; meanwhile the size of samples in each direction is fixed. This process is mainly to optimize the local structure, specifically the interface structure here. Hence, due to size limitations, there are still some residual stresses in all directions after energy minimization. To relax the internal residual stress, samples are relaxed at 300 K for 20 ps. In the relaxation, to relax the stress in all directions to zero pressure, the sample size is allowed to change.

Point 16: Line 178 “atomic fractions of different local” how to find it. Explain in detail. Figure 4a, why σ dropped in stage 2 correlate this with stage I and stage III. Figure 4b, why atomic fraction of HCP is lower than that of FCC and what is the reason for the change in stage 2 and parallel behavior Figure 4c.

Response 16: Thank the reviewer for this kind comment.  Using the DXA algorithm, we can classify atoms into different classes based on local microstructures, and then calculate the number of atoms in different local structures to get their atomic fractions. The stress drop in the second stage is because the stress reaches the critical stress of dislocation nucleation, which induces the yield of the material and the nucleation of dislocations.

 “atomic fraction of HCP is lower than that of FCC”:  This is mainly because the FCC is the main crystal structure, and dislocation nucleation and slip-induced microstructural changes are not sufficient to convert all FCC structures to HCP structures.

The reason for the change in stage 2 and parallel behavior Figure 4c:  In Stage 2, it includes dislocation nucleation and the reaction between them. A large number of dislocations form, which brings in the change of the curves; however,  the number of dislocations did not change much during the reaction stage.

The corresponding analysis has been added to the revised manuscript.

Point 17: Why there is no change in stage 3 and 1/6<112> exhibited lower value COC and twin Figure 4d. the 1/6<110> twin displays higher value while 1/6<112> shows lower value. What is the reason for and why <110> shows lower value than <112>

Response 17: Thank the reviewer for this kind comment. In Stage (III), the reaction of dislocations is the primary behavior, and the reactions that occur are mainly the decomposition of <110> perfect dislocations into <112> partial dislocations, or the synthesis of <112> partial dislocations into <110> perfect dislocations. Therefore, two kinds of dislocations number of changes in the law are reversed.

The corresponding analysis has been added to the revised manuscript.

Point 18: What is the different between fig 4a and 7. Author is advised to include reflective planes of <110> or <112> in figure 8.

Response 18: Thank the reviewer for this kind suggestion. The difference between Fig. 4a and 7 is that,  in Fig. 7, the σε curves for different modulation periods are presented. In Fig. 8, it does not involve the number and density of different dislocations (<110> or <112>), hence these were not added in the revised manuscript.

Point 19:  Make conclusion into a paragraph.

Response 19: Thank the reviewer for this kind suggestion. We have merged the conclusion into a paragraph in the revised manuscript.

Reviewer 2 Report

In this manuscript the authors report an interesting analysis to investigate the effects of interfacial structure and modulation period of Cu/Pd multilayered films by means of molecular dynamic simulations. I think that the topic of manuscript is noteworthy of investigation since it provides improvements in the field. I suggest to publish this research on Nanomaterials after minor remarks:

Both in the abstract and in the introduction part the authors should better emphatize the aim of the work. Please, revise reference list; some abbreviation are missing.

Author Response

Response to Reviewer 2 Comments

Point 1:  In this manuscript, the authors report an interesting analysis to investigate the effects of interfacial structure and modulation period of Cu/Pd multilayered films by means of molecular dynamics simulations. I think that the topic of the manuscript is noteworthy of investigation since it provides improvements in the field. I suggest to publish this research on Nanomaterials after minor remarks:

Response 1: We thank the reviewer very much for valuing our manuscript and for his/her recommendation of our manuscript in the possible publication in Nanomaterials.

Point 2: Both in the abstract and in the introduction part the authors should better emphasize the aim of the work.

Response 2: Thank the reviewer for this kind suggestion. We have added descriptions in the abstract and introduction to emphasize the purpose of this article's work.

Point 3:  Please, revise reference list; some abbreviation are missing.

Response 3: Thank the reviewer for this kind suggestion. We have checked and modified the reference format to avoid missing information as well.

Round 2

Reviewer 1 Report

Accept in current form